# An Integrated Approach for the Determination of Young's Modulus of a Cantilever Beam Using Finite Element Analysis and the Digital Image Correlation (DIC) Technique

**Tick Boon Loh** [1,*] , **Yutong Wu** [2] , **Siang Huat Goh** [2] , **Kian Hau Kong** [2] , **Kheng Lim Goh** [1] and **Jun Jie Chong** [1]

1 Faculty of Science, Agriculture & Engineering, Newcastle University in Singapore, SIT Building @ Nanyang Polytechnic, 172A Ang Mo Kio Avenue 8 #05-01, Singapore 567739, Singapore
2 Department of Civil and Environmental Engineering, National University of Singapore, 21 Lower Kent Ridge Road, Singapore 119077, Singapore
* Correspondence: t.b.loh2@newcastle.ac.uk

**Abstract:** This paper is an extended paper from the 24th International Conference on Mechatronics Technology, ICMT 2021. The basic mechanical characteristic that gauges the stiffness of a solid material is known as the Young's modulus. To evaluate the Young's modulus, destructive material testing is frequently used. This paper describes how to determine a material's dynamic Young's modulus using Digital Image Correlation (DIC) in conjunction with numerical back-analysis. Three different materials (brass, aluminum, and steel) were examined for their static and dynamic reactions. A static transverse displacement was first applied at the free end of the beam before it was released and the beam was allowed to vibrate freely. The resulting vibrations at the free end were monitored using the DIC method, following which the natural frequencies of the beam were derived by applying the Fast Fourier Transform (FFT) to the DIC measured time history. The Young's modulus corresponding to the fundamental natural frequency of the beam was then obtained via modal back-analysis using the finite element program Ansys 2022 R1. In this way, the Young's modulus of the material may be calculated using a combination of numerical and DIC techniques, thus allowing for the non-contact evaluation of the structural integrity without subjecting the material to destructive testing. Potential applications of this method include bridge and building assessments, and structural health monitoring (SHM).

**Keywords:** digital image correlation; static and dynamic Young's modulus; cantilever beam; natural frequency; numerical

## 1. Introduction

Young's modulus, also known as elastic modulus *E*, is a mechanical characteristic that reflects a material's stiffness [1]. Young's modulus is crucial in describing material reactions since it shows how a structural element's stress and strain are related. Its value could be measured using a variety of procedures and test methods. These techniques are often divided into static and dynamic techniques [2].

Static techniques are usually based on the linear slope calculated in the linear regime of the stress-strain curve, which is determined by direct measurement [2]. The standard tensile, compression, or flexural tests are among the static methods commonly adopted [3–5] These techniques are typically destructive as the test specimens are subjected to increasing deformation until they break. Such static approaches are more often used by engineers than dynamic modulus methods. To determine sufficiently accurate values of the Young's modulus, some of the more important practical concerns relating to the tensile test are addressed and explained in [6].

Compared with static methods, dynamic methods offer advantages such as higher accuracy, compatibility with objects of varying forms and sizes, non-destructive measure-

ment, and testing in various environmental conditions [3,7–9]. These techniques involve testing at low strain levels and measuring the test specimens' natural frequencies when they are loaded by excitation at or near resonance [8]. Since the elastic limit of the material response is not reached at lower strain levels, this approach is essentially non-destructive and permits repeated testing on the same specimens [3,7].

Studies have been reported in the literature, which study how static and dynamic Young's modulus differ in terms of values and applications [3,7]. These investigations have demonstrated that dynamic approaches have higher accuracy, with errors ranging from $\pm 1$ to 2%. The research in [10] shows the distinction between dynamic and static Young's moduli for typical austenitic steel. Due to the challenges involved with using static methods to obtain high-quality experimental findings, dynamic methods are widely utilized for the measurements of ceramic materials [11]. The investigation in [12] shows the statistical relationship between the static and dynamic Young's moduli of rock materials. While Young's modulus calculated using dynamic techniques ($E_d$) is higher than the modulus calculated using static methods ($E_s$), studies have shown that the dynamic method result is actually 2% lower than the static method result and falls within the 6% standard deviation of the results calculated from static methods [10,13].

Due to the difficulty of obtaining test materials for static studies in the laboratory, the dynamic Young's modulus can frequently be determined using logging data obtained in the field. Additionally, newer methods have been devised to calculate the dynamic Young's modulus more quickly and without the use of extra equipment to stimulate resonance conditions [14]. However, the majority of these studies use traditional contact-based measuring methods. With rapid technological advances, new non-contact optical techniques have been proposed and developed into measurement instruments, thus providing opportunities for employing such approaches to do dynamic testing.

Digital image correlation (DIC) is proposed as an alternative technique for dynamic testing. It is a non-contact optical method of measurement that uses digital image processing and numerical computing [15–17] to derive displacement and deformation. The DIC method was initially developed in the 1980s [18,19] and has since been improved as a result of the availability of higher-quality digital photos and significant algorithmic advancements [20]. These developments have made possible the higher acquisition rate of capturing images during testing, thus deriving detailed and precise measurements during postprocessing [21]. The advantages have led to DIC becoming a highly important tool in the experimental mechanics community [22].

DIC is one of the most popular non-destructive techniques for automatically calculating full-field strains on rock specimens [23]. In [23], DIC was used in the study of multi-scale strain-field heterogeneity in rocks. As it doesn't require direct access to or contact with the structure, it is also a cost-effective technique for structural health monitoring (SHM).

An essential tool for examining a structure's dynamic properties is modal analysis. It creates a set of modal parameters based on the vibration excitation and response signals. Modal analysis utilizing a finite element analysis technique was used to determine the natural frequencies and frequency response of nettle/polyester and chicken feathers [24]. It is suggested in [25] that DIC can be utilized to quantify full-field displacements, together with finite element simulations, to determine the pace at which surface fractures release strain energy.

In this study, three metallic materials (steels, aluminum, and brass) were subjected to dynamic excitations, and the free vibration responses were captured using full-field 2D digital image correlation (2D-DIC). To determine the Young's modulus of these materials, this paper combines the use of DIC with numerical modal analysis using the finite element program Ansys.

## 2. Theoretical Background

### 2.1. Euler–Bernoulli Cantilever Beam Theory

Figure 1 shows a uniform cantilever beam with one end clamped to provide a fixed support condition. The tested beam has a high length-to-thickness ratio ($l/t$) of 182 ($l \gg$ t), so that the shear effects are minimized and may be ignored. According to ASTM standards [26], a span length to thickness ratio of 24 is recommended. The differential equation [27] based on Euler–Bernoulli beam theory may thus be used to compute the flexural displacement $y(x)$ under a static point load $P$ applied at the free end as follows:

$$\frac{\partial}{\partial x^2}\left[EI\frac{\partial^2 y(x)}{\partial x^2}\right] = 0 \qquad 0 < x < l \tag{1}$$

with boundary conditions as follows:

$$y(0) = 0 \quad \text{and} \quad \frac{\partial y}{\partial x}(0) \quad at\ x = 0 \tag{2}$$

$$EI\frac{\partial^2 y}{\partial x^2}(l) = 0 \quad \text{and} \quad \frac{\partial}{\partial x^2}\left[EI\frac{\partial^2 y}{\partial x^2}(l)\right] = -P \quad at\ x = l \tag{3}$$

The maximum deflection $\delta$ is given by the following:

$$\delta = \frac{Pl^3}{3EI} \tag{4}$$

where $l$ is the cantilever beam's gauge length, $E$ is the material's Young's modulus, and $I$ is the area moment of inertia for the beam's cross-section. Equation (4) may be re-arranged to find the static Young's modulus $E_s$.

$$E_s = \frac{-Pl^3}{3\delta I} \tag{5}$$

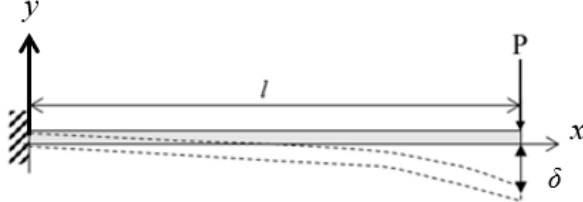

**Figure 1.** A uniform fixed-end cantilever beam with a static point load P supplied at the free end.

### 2.2. Cantilever Beam Vibration Equation

The Euler–Bernoulli cantilever beam equation may be used to obtain the natural frequency $f_n$ ($s^{-1}$) of the first mode of vibration when the beam is subjected to free vibration [27] as follows:

$$f_n = \frac{K_n}{2\pi}\sqrt{\frac{EIg}{wl^4}}, \quad n = 1,\ 2,\ 3\ldots \tag{6}$$

where the eigenvalue constant $K_n$, the vibration mode $n$, the gravitational acceleration constant $g$, and the uniform load per unit length $w$ are all defined.

Equation (3) can be rewritten as follows to yield the dynamic Young's modulus $E_d$ as follows:

$$E_d = \frac{wl^4}{Ig}\left(\frac{f_n 2\pi}{K_n}\right)^2 \tag{7}$$

### 3. Experimental Setup

*3.1. 2D-Digital Image Correlation (2D-DIC) System*

The basic principle of DIC is to correlate two images taken from a specimen surface before and after deformation. The first image will be the reference image, and the second image will be the deformed image [28]. It is assumed that the features of the object's surface are being displaced together with the object. As a result, an area where the grey-level distribution after deformation is the same as the grey-level distribution of the subset before deformation is being sought [17]. Through the comparison of the subsets of the images, mathematical mapping and cross-correlation can be carried out to obtain the displacement field [28].

In this study, the beam's vibrations were measured using the industrial DIC system developed by Imetrum. The monochrome camera (IM-Cam-033-UVX) with 25 mm lenses is attached to a computer processing unit as the hardware component. The Video Gauge software application was used to control, record, process, and evaluate the non-contact measurements.

Video Gauge is a software developed based on digital image correlation techniques supplied by Imetrum. The software allows fast analysis of results, suitable time synchronization with other measurement sensors as well as ease of usage [29]. Research performed by [30] has shown that the Video Gauge software is able to track unmarked surfaces accurately. Compared to strain gauges, the results from the DIC lenses with different fields of view have displayed good correlations. The accuracy of the displacement measurements obtained using Video Gauge was also analyzed by [31]. The measurement uncertainty increases with the increment of working distance and decreases as the focal length reduces. During the calibration process, a measuring ruler was placed on the beam to specify the actual length of 50 mm between 2 defined points in an image. It is observed that the measurement uncertainty will also decrease when the length of the two defined points increases. As such, the error bounds can be taken into consideration when conducting the experiments, and proper calibration conducted prior to the test.

A stochastic pattern, such as white dots on a black backdrop or black dots on a white background, must be present on the surface of interest in order to perform DIC measurements. Certain patterns will produce higher tracking resolution, while other types are more appropriate for specific purposes. The ideal target patterns should have a range of tones of gray, including both light and dark parts. The actual size of the target pattern is dependent on the field of view (FOV) and the camera resolution. The default target size is $80 \times 80$ pixels in the software. The common types of patterns are speckles, blobs, dashes, concentric rings, and natural features [32]. In this experiment, blob patterns were hand-drawn at the ends of cantilever beam surfaces using a marker pen.

To record the displacements and dynamic response of the cantilever beam during vibration, the camera was configured to capture images at a frame rate of 66 Hz with a resolution of $728 \times 533$ pixels and a pixel size of 0.0069 mm. The camera was rigidly attached to a camera tripod on firm ground to prevent unnecessary movements. It was also connected to a portable computer using an Ethernet cable. As seen in Figure 2, the distance between the camera and the target site was set as 1 m, and the target location was set as the free end of the beam.

*3.2. Materials*

The test specimens are made up of three rectangular beams, each composed of a different material. These materials are steel, aluminum, and brass. All specimens have the same dimensions as follows: 19 mm in width, 3.3 mm in thickness, and 650 mm in total length. All specimens have an area moment of inertia $I$ of $5.69 \times 10^{-11}$ m$^4$. The densities of the materials shown in Table 1 were determined by dividing the measured mass of the beam by its volume. The material is assumed to be isotropic.

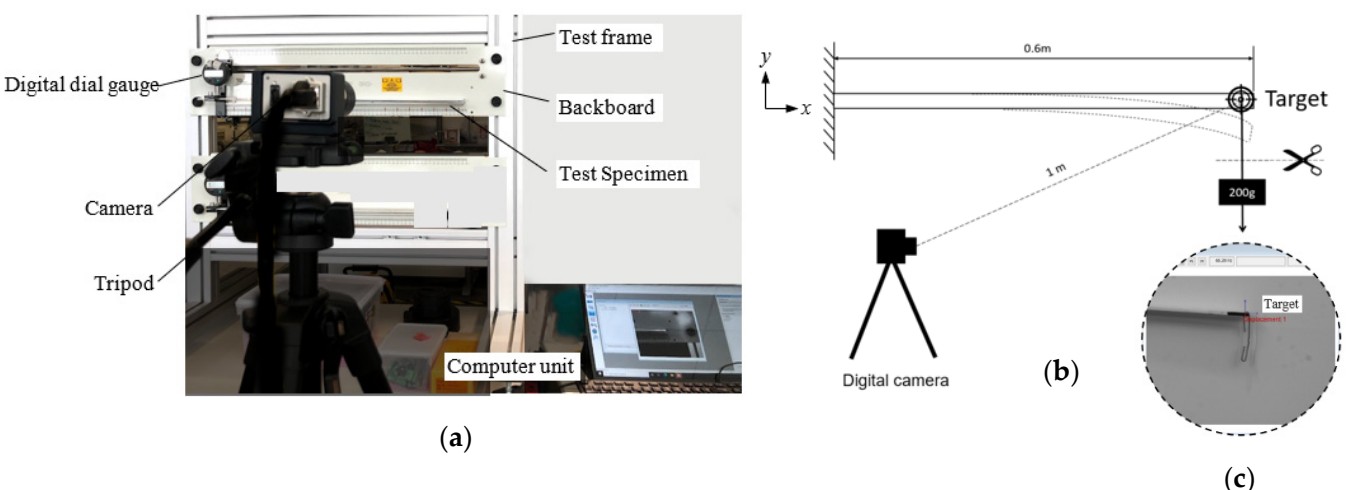

**Figure 2.** (**a**) Equipment setup presenting the Cantilever Beam Test Frame as measured by the DIC system, (**b**) A diagram of the equipment's setup displaying the camera's position, (**c**) the field of view, and a target that Video Gauge software has recorded.

**Table 1.** Density of materials.

| Materials | Steel | Aluminium | Brass |
|---|---|---|---|
| Density [kg/m$^3$] | 8195 | 2700 | 8170 |

*3.3. Experimental Analysis*

As shown in Figure 2, the experiment was carried out using a backboard mounted to a rigid aluminum test frame. All the test beams had a total length ($L$) of 650 mm. At one end, the specimens were clamped horizontally across a 50 mm gap. The effective beam length ($l$) is thus 600 mm, which is obtained by subtracting the 50 mm grip length from the overall length ($L$). A plate and screw assembly was used to clamp each specimen at its fixed end ($x = 0$), with the screw being tightened to a consistent tension to hold the specimen in place. As shown in Figure 2, the specimen can freely deflect in the y-direction.

At the cantilever's free end, a 200 g mass was hung. At the free end of the specimen, a digital dial gauge that measures vertical deflections was installed in order to validate the displacements measured by DIC during the initial static loading. This static displacement measurement can be used to derive the static Young's modulus $E_s$ using Equation (5) and the results compared with the Ansys back-analysis of the same problem.

The 200 g mass was then instantaneously released from the end of the specimen, thus causing the beam to undergo free vibrations. The time history of the vibrations at the free end of the beam was captured by the DIC system in the form of a digital image response. By applying fast Fourier transform (FFT) to the measured time histories processed from the DIC images, the specimen response in the frequency domain can be obtained as shown in Figure 3 for the brass beam, with the spike showing the natural frequency $f_n$ of the free vibrations. Equation (7) can then be used to back-calculate the dynamic Young's modulus $E_d$ of the beam material, and this value of $E_d$ can then be compared with Ansys modal analysis results. The DIC experiment was repeated three times for each specimen to check for consistency and repeatability of the results.

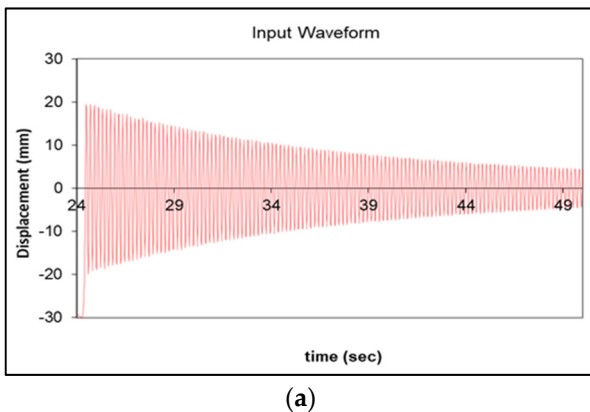
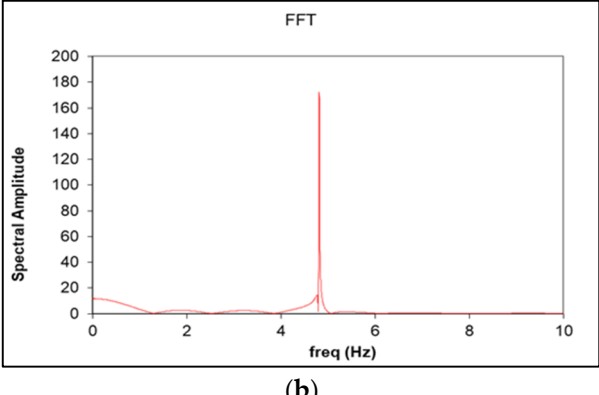

<div align="center">(<b>a</b>)              (<b>b</b>)</div>

**Figure 3.** (**a**) The vibration response within the time domain for the brass specimen, (**b**) FFT displaying the 4.8 Hz natural frequency that was determined.

### 3.4. Numerical Analysis

Finite element modal analysis using Ansys 2022R1 software [33] is used to determine the natural frequencies and mode shapes of the cantilever beam due to the free vibrational response generated by the release of the 200 g static load. The geometric model shown in Figure 4a was created using Ansys SpaceClaim. The dimensions of the model are 600 mm in length, 19 mm in width, and 3.3 mm in thickness. The prescribed material properties of the three materials, steel, aluminum, and brass, are shown in Table 2, where the Young's modulus *E* is allowed to vary within a range. Eight-node brick elements are used to create the uniform 3D mesh. Each brick element size is 1 mm. In total, the 3D model contains 34,200 elements and 177,816 nodes. The meshed model has a fixed support constraint at one end. The other end is loaded with a point load of 1.962 N. The static deflection $\delta$ due to the point load is calculated using the Ansys structural package. Separately, the Ansys modal analysis package was used to calculate the natural frequencies ($f_n$) of the 3D discretized model under different vibration modes, as shown in Figure 4b.

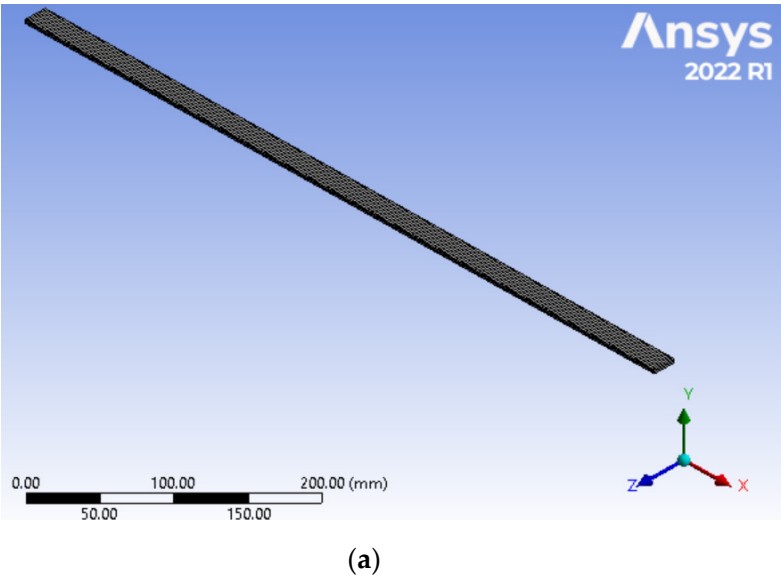

<div align="center">(<b>a</b>)</div>

**Figure 4.** *Cont.*

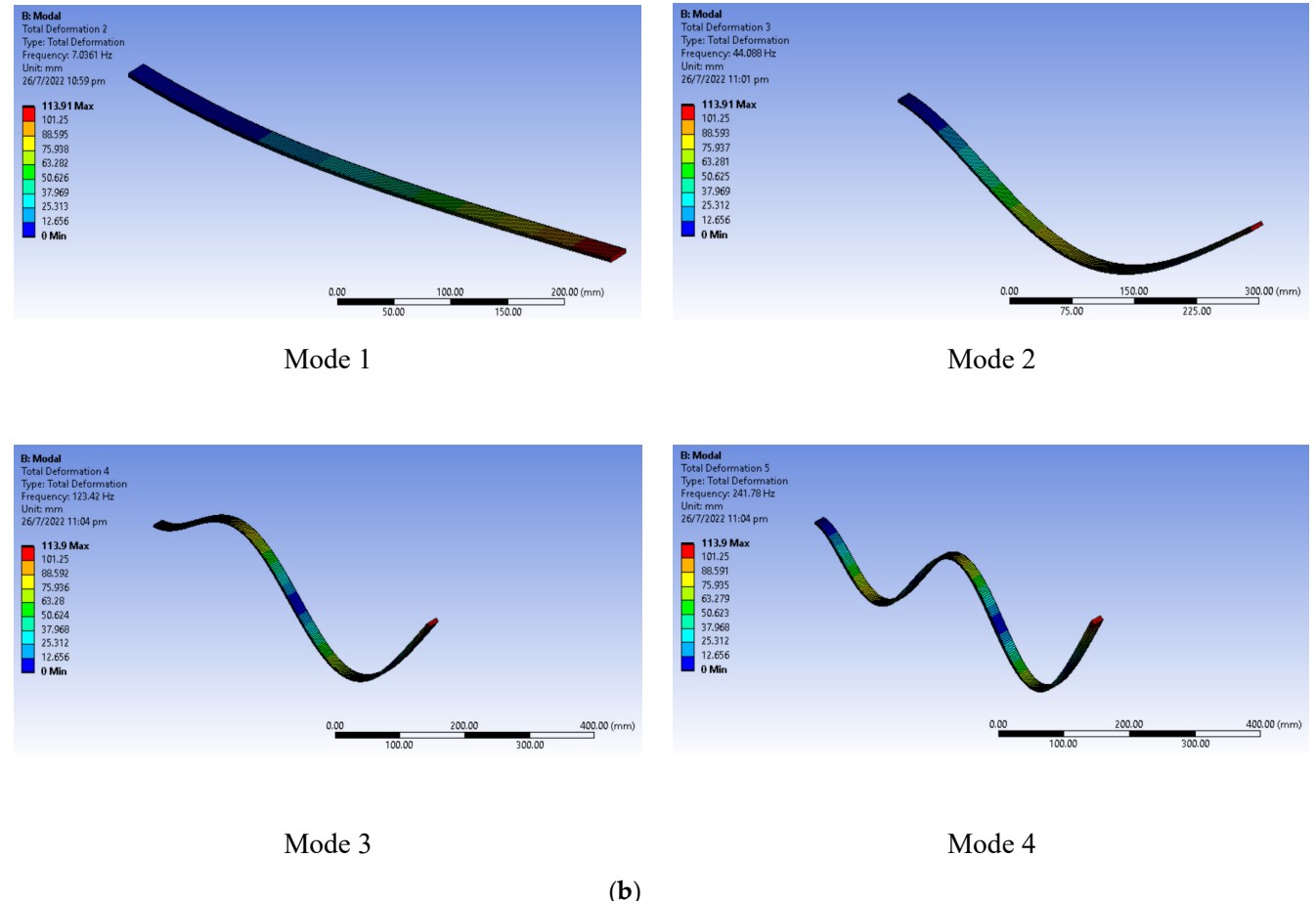

Mode 1

Mode 2

Mode 3

Mode 4

(**b**)

**Figure 4.** (**a**) Geometry model of cantilever beam, (**b**) modal analysis of natural frequencies ($f_n$) under different vibration modes.

**Table 2.** Physical properties of assigned materials.

| Materials | Young's Modulus (*E*) Range [GPa] | Poisson's Ratio |
|---|---|---|
| Steel | 170 to 200 | 0.3 |
| Aluminum | 60 to 90 | 0.3 |
| Brass | 60 to 90 | 0.3 |

## 4. Results and Discussion

### 4.1. A. Comparison of Measurements Using DIC and Accelerometer

An accelerometer was fastened to the steel specimen's free end to measure the accelerations of the beam's free vibrational response. The findings are displayed in Figure 5a, which shows the displacements (in mm) measured by the DIC and the accelerations (in m/s$^2$) measured by the accelerometer on the same horizontal time axis but different vertical axes. It is noted that there is an obvious similarity and correlation between the acceleration histories recorded by the accelerometer and the DIC-derived displacement time histories. This comparison shows that the DIC approach was able to accurately capture the signal trend since the peak values occurred at the same time in both sets of data. When comparing the peak spectral amplitude in the frequency domain, Figure 5b also displays the same signal trend. The FFT results show that the natural frequency is 6.45 Hz as recorded by the accelerometer and 6.38 Hz as calculated using the DIC-derived displacements. This shows that the DIC approach can accurately capture the beam's vibrational response and generate frequency contents that are almost identical to those obtained using the accelerometer.

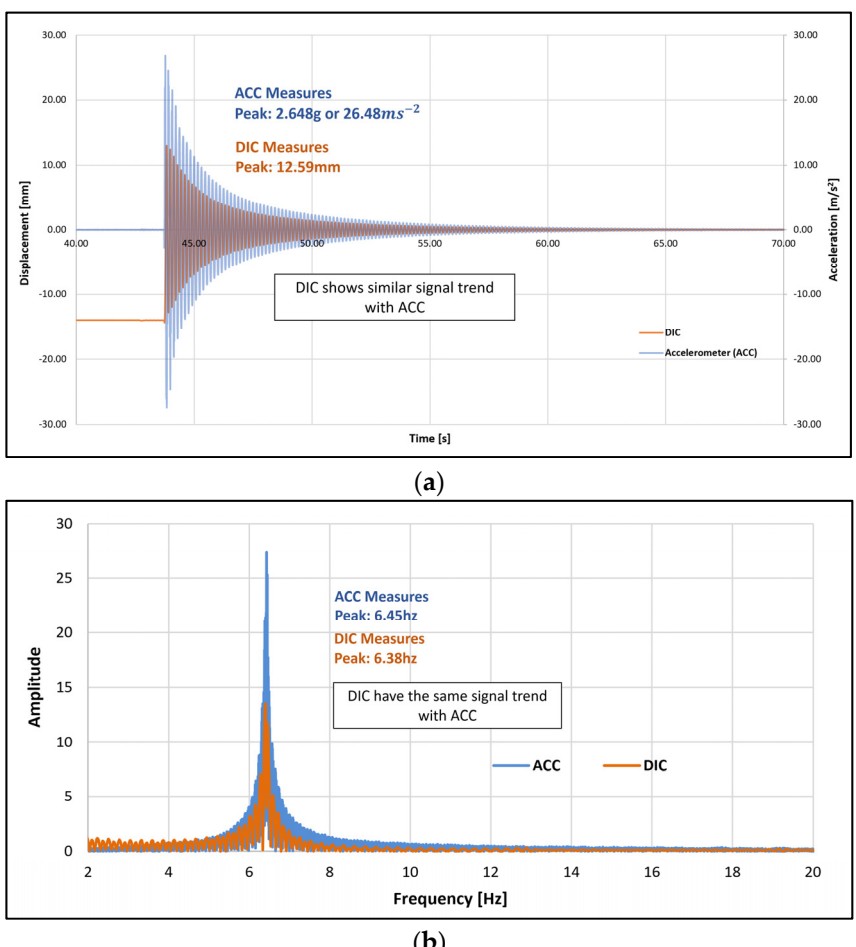

**Figure 5.** Comparison of (**a**) the free vibration frequency measurement using the accelerometer and the DIC, (**b**) the FFT amplitude peak values of the accelerometer and the DIC.

### 4.2. B. Validation of Numerical with Theoretical Results

Table II shows the numerically computed static deflection ($\delta$) and natural frequency ($f_n$) of the steel beam based on different Young's modulus $E$ (from 170 to 200 GPa) compared with the corresponding theoretical values calculated from Equations (4) and (7). The close agreement in the results affirms the accuracy of the numerical solutions. The percentage errors shown in columns 4 and 7 of Table 3 pertain to the static deflection values and the natural frequencies of the beam, respectively. The values presented in these two columns are not errors in the Young's modulus $E$. What the results show is that, for any given Young's modulus $E$, the errors in the static deflections computed using theory and FEM are very small, between 0.01% and 0.015%. For any given Young's modulus $E$, the errors in the natural frequencies computed using theory and FEM are also very small, ranging from 0.09% to 0.1%. The reason why the errors in the computed natural frequencies are higher than the computed static deflections is likely related to the different numerical techniques used to calculate these two quantities. The static deflections are calculated using conventional stress/deformation-based techniques, while the natural frequencies are calculated using modal techniques that involve eigenvalue analysis. In this study, the eigenvalue analyses result in higher percentage errors compared to the stress/deformation analyses. However, in both cases, the percentage errors are overall very small, and hence the FEM values can be considered to show good agreement with the theoretical values.

**Table 3.** Static deflection and natural frequency values of steel as a function of E calculated using theoretical and Ansys numerical analysis.

| Young's Modulus (*E*) [GPa] | Deflection (*δ*) [mm] | | Error [%] | Natural Frequency (*f*$_n$) [Hz] | | Error [%] |
|---|---|---|---|---|---|---|
| | Theory | FEM | | Theory | FEM | |
| 170 | 14.604 | 14.602 | 0.014% | 6.751 | 6.745 | 0.094% |
| 175 | 14.187 | 14.185 | 0.014% | 6.850 | 6.843 | 0.093% |
| 180 | 13.793 | 13.791 | 0.015% | 6.947 | 6.940 | 0.095% |
| 185 | 13.420 | 13.418 | 0.015% | 7.043 | 7.036 | 0.094% |
| 190 | 13.067 | 13.065 | 0.015% | 7.137 | 7.131 | 0.094% |
| 195 | 12.732 | 12.730 | 0.016% | 7.231 | 7.224 | 0.094% |
| 200 | 12.413 | 12.412 | 0.008% | 7.323 | 7.316 | 0.095% |

*4.3. C. Nummerical Results of Static and Dynamic Young's Moduli from DIC Measurements*

This section discusses the determination of static and dynamic Young's moduli by performing back-analysis using the Ansys finite element to match the DIC results for static deflection and natural frequency, respectively. Table 3 shows three sets of DIC measurements obtained from tests conducted using steel, aluminum, and brass beams. The static Young's modulus $E_s$ reported in Column 4 are the input values in the Ansys program that yield the closest agreement between the static deflections obtained from DIC measurements (Column 2) and those computed by the FEM stress/deformation analysis. The dynamic Young's modulus $E_d$ reported in Column 5 are the input values in the Ansys program that result in the closest agreement between the natural frequencies obtained from the DIC measurements (Column 3) and those computed by the FEM modal analysis.

As expected, steel has the highest static and dynamic Young's modulus values of 175.5 GPa and 186.5 GPa, respectively. The difference between the static and dynamic modulus is about 6%. It has the smallest deflection of 14.162 mm and the lowest natural frequency of 7.0 Hz. On the other hand, aluminum has the lowest static and dynamic Young's modulus values of 66.7 GPa and 63.8 GPa, respectively, with the difference being about 4.3%. It has the largest deflection of 37.214 mm and the highest natural frequency of 7.2 Hz. The static and dynamic Young's modulus of brass are obtained as 82.0 GPa and 86.3 GPa, respectively, the difference being about 5%. While the magnitude of the static deflection of the brass beam is between that of steel and brass, its natural frequency is the lowest among the three. This is because, unlike deflection, which is related chiefly to the material modulus, the natural frequency is affected by both the material modulus and the material density. Overall, the results in Table 4 show that, for each of the materials tested, the difference between the static and dynamic Young's modulus is generally from 4% to 6%. The summary of steel, aluminum, and brass' static and dynamic Young's moduli and natural frequencies is shown in Figure 6.

**Table 4.** Comparison of static and dynamic Young's moduli of steel, aluminum, and brass.

| Materials | DIC Measurement | | Young's Modulus (from FEM Back-Analysis to Match DIC Results) | | |
|---|---|---|---|---|---|
| | Static Deflection (*δ*) [mm] | Natural Frequency (*f*$_n$) [Hz] | Static Modulus (*E*$_s$) [GPa] | Dynamic Modulus (*E*$_d$) [GPa] | Difference [%] |
| Steel | 14.162 | 7.0 | 175.5 | 186.5 | 5.90 |
| Aluminum | 37.214 | 7.2 | 66.7 | 63.8 | 4.34 |
| Brass | 30.251 | 4.8 | 82.0 | 86.3 | 4.98 |

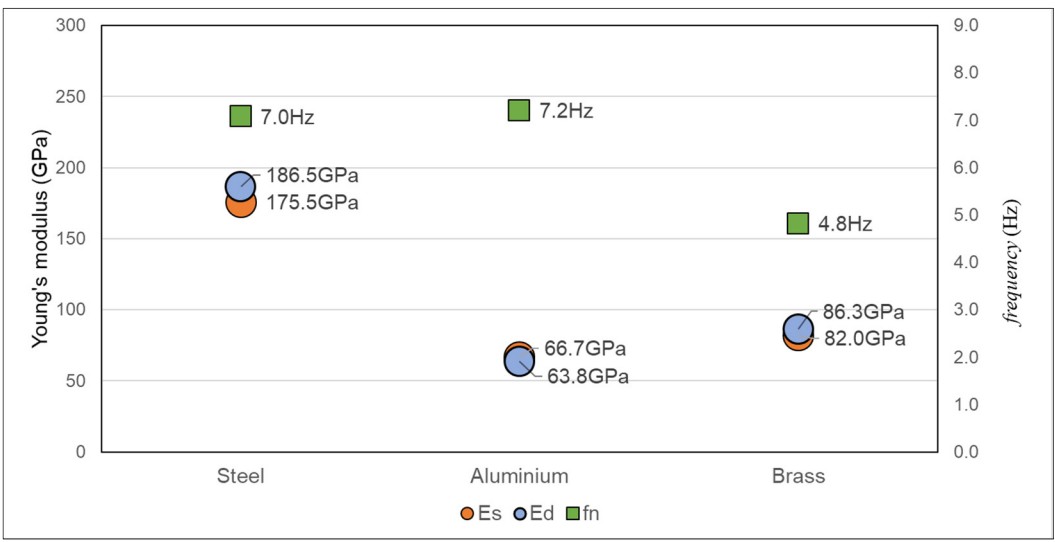

**Figure 6.** Summary of static and dynamic Young's moduli and natural frequency of steel, aluminium and brass.

## 5. Conclusions

This study illustrates the combination of digital image correlation (DIC) and the finite element method as an alternative non-contact methodology for evaluating the Young's modulus of materials. The following is a summary of the findings:

- The contact-based accelerometer measurements are used to provide a check on the reasonableness of the trends measured by the contactless DIC method. The results demonstrate good agreement between the two measuring techniques in terms of time-history trends and the natural frequencies;
- The vibration response of the cantilever beam can be measured using the contactless DIC technique. By performing back-analysis using a finite element program to match the DIC response, the static and dynamic Young's modulus of the beam can be obtained;
- For the steel, aluminum, and brass materials tested, the difference between the static and dynamic Young's moduli is generally between 4% and 6%;
- By applying the Fast Fourier Transform (FFT) to the DIC measured displacement histories, the natural frequencies of the beam can be obtained. The experiments show that the non-contact DIC methodology could be used to obtain a structure's inherent frequencies instead of the more traditional contact-based approaches.

Cantilever beam tests are often performed in controlled laboratory environments, but field situations are far more complicated. Therefore, more research and analysis are required to establish the viability of employing the DIC approach for evaluating composite materials such as reinforced concrete in real-world conditions outside of the laboratory. This could open the door to using DIC as a low-cost, contactless method for assessing structural health.

**Author Contributions:** Conceptualization, T.B.L. and K.L.G.; methodology, T.B.L. and S.H.G.; software, K.H.K.; validation, T.B.L., Y.W. and S.H.G.; formal analysis, T.B.L., Y.W., S.H.G. and K.H.K.; investigation, T.B.L. and Y.W.; resources, J.J.C., K.L.G., S.H.G. and K.H.K.; data curation, T.B.L.; writing—original draft preparation, T.B.L. and Y.W.; writing—review and editing, T.B.L., Y.W. and S.H.G.; visualization, T.B.L.; supervision, K.L.G., J.J.C. and S.H.G.; project administration, J.J.C.; funding acquisition, J.J.C. All authors have read and agreed to the published version of the manuscript.

**Funding:** This research received no external funding.

**Institutional Review Board Statement:** Not applicable.

**Informed Consent Statement:** Not applicable.

**Data Availability Statement:** No new data were created or analyzed in this study. Data sharing is not applicable to this article.

**Acknowledgments:** The authors would like to thank The Land Transport Authority of Singapore, Republic Polytechnic, Sensorcraft Technology, and Imetrum Ltd., for their support.

**Conflicts of Interest:** The authors declare no conflict of interest.

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
