# Peer review of "An Integrated Approach for the Determination of Young’s Modulus of a Cantilever Beam Using Finite Element Analysis and the Digital Image Correlation (DIC) Technique"

_electronics, doi:10.3390/electronics11182826_

Round 1

Reviewer 1 Report

1. The induction seems to be fragmented, please consider to reorganize it.

2. Page 2, line 53 “studies have been conducted to investigated….” should be changed to “tudies have been conducted to investigate…”

Line 56: hat does “In [Parasites]” mean?

3. Equations seemed not well formatted, consider improving them?

4. The description of DIC setup is too concise, a lot of info needs to be added, such as the speckle setting, image noise evaluations, which the authors may refer to https://doi.org/10.1016/j.ijrmms.2020.104212 and https://doi.org/10.1016/j.tafmec.2021.103014 for more details.

5. It seems to me 2D DIC was applied but FEM model was 3D, will this be conflict? Or will the results comparable?

6. How did the authors ensure that there was no out-of plane movement during image acquisition?

7. A list of parameters is suggested to add for the ANSYS simulations. 

Author Response

No.

Comments and Response

1

The induction seems to be fragmented, please consider to and Suggestions for Authors

reorganize it.

Please accept our apologies for the earlier version of the manuscript which you reviewed.  We had mistakenly submitted an unedited version of the manuscript for review, which we later realize is full of language and grammatical errors.  We are sincerely sorry for the frustration you had experienced in trying to review and understand the contents of the unedited manuscript.  Your patience and understanding in reviewing the paper, despite the major language issues, is greatly appreciated.

The revised manuscript has been corrected and carefully checked to minimize such language issues.

2

Page 2, line 53 “studies have been conducted to investigate ….” should be changed to “tudies have beenconducted to investigate…” Line 56: hat does “In [Parasites]” mean?

The phrase “In [Parasites]” should not be there.  It has been removed in the revised manuscript.

3

Equations seemed not well formatted, consider improving them?

The equations have been properly formatted in the revised manuscript.

4

The description of DIC setup is too concise, a lot of info needs to be added, such as the speckle setting, image noise evaluations, which the authors may refer to

https://doi.org/10.1016/j.ijrmms.2020.104212 and https://doi.org/10.1016/j.tafmec.2021.103014 for more details.

We thank the reviewer for the recommended publications.  We have included more details of the DIC setup in the revised manuscript, as advised by the reviewer.

5

It seems to me 2D DIC was applied but FEM model was 3D, will this be conflict? Or will the results comparable?

There will be no conflict when comparing the results from the 2D DIC experiment with those computed from a 3D FEM model. 

This is because the beam deflections measured by the DIC takes place fully within the vertical plane.  The 3D FEM model was also subjected to a loading such that the beam deflections also take place within the same vertical plane, without any horizontal or torsional effects.

Hence, the results from the 2D DIC experiments and the 3D FEM model can be compared against each other, based on the way that the beam is loaded.

6

How did the authors ensure that there was no out of plane movement during image acquisition?

This experiment involves the vertical (transverse) deflection of a slender beam, due to a transverse point loading applied at the free end of the beam.  Such a loading will result in almost negligible out-of-plane movement of the beam.

Our present experiment is different from the case of a concrete cube or cylindrical rock sample subjected to an axial loading, which will generate significant out-of-plane movements due to the Poisson’s ratio effect.

7

A list of parameters is suggested to add for the ANSYS simulations

We thank the reviewer for this suggestion, and have included the list of parameters adopted in the ANSYS simulations in the revised manuscript.

Reviewer 2 Report

The comments of reviewer on electronics-1868638 (score: 80/100):

This study is based on the determination of Young’s modulus in cantilevered beams using digital image correlation (DIC). Using the Euler-Bernoulli beam model, the static lateral deflection of the tip of the beam as well as its natural frequencies are theoretically displayed and based on the measurements via the DIC. The elastic modulus values for cantilevered beams of various materials are estimated based on the abovementioned static and dynamic approaches and verified, and then the capabilities of the DIC in capturing the theoretical values of the Young’s modulus are displayed and discussed. The paper can be accepted for publication in Electronic-MDPI after the following major issues are rationally explained and replied by the authors:

1.       According to Table II, it seems that the numerically evaluated Elastic modulus values based on the natural frequency (i.e., dynamic analysis) have higher relative errors with respect to those evaluated based on the static-based analysis. Why? Please clarify within the paper manuscript.

2.       Under what circumferences, we can use Eq. (1) for static analysis of beam-like structures? Please clarify carefully within the manuscript of the paper.

3.       For beams with length to height ratio lower than 5, the role of the shear deformation in the maximum deflection and natural frequencies of cantilevered beams become important. Is the proposed approach by the authors would be also applicable to these special but crucial cases as well? How? Please explain with some detail.

4.       In view of comment#3, the literature review on the application of shear deformable beam theories to the statics and dynamics of beam-like structures is somehow NOT complete! The authors are invited to discuss on the following works that employ Timoshenko and higher-order beam models for structural analysis of the beam-like structures:

(*) Zhang K, Ge MH, Zhao C, Deng ZC, Xu XJ. Free vibration of nonlocal Timoshenko beams made of functionally graded materials by Symplectic method. Composites Part B: Engineering. 2019 Jan 1;156:174-84.

(*) Pinnola FP, Barretta R, Marotti de Sciarra F, Pirrotta A. Analytical solutions of viscoelastic nonlocal timoshenko beams. Mathematics. 2022 Feb 1;10(3):477.

(*) Aichun, L. and Kiani, K., 2020. Bilaterally flexural vibrations and instabilities of moving piezoelectric nanowires with surface effect. The European Physical Journal Plus, 135(2), pp.1-29.

(*) Yu G, Kiani K, Roshan M. Dynamic analysis of multiple-nanobeam-systems acted upon by multiple moving nanoparticles accounting for nonlocality, lag, and lateral inertia. Applied Mathematical Modelling. 2022 Aug 1;108:326-54.

(*) Mu B, Kiani K. Surface and shear effects on spatial buckling of initially twisted nanowires. Engineering Analysis with Boundary Elements. 2022 Oct 1;143:207-18.

5.       Are there other effective non-destructive tests for measuring the elastic modulus of beams? Please clarify.

6.       There are some typos and badly constructed sentences that should be carefully corrected. For instance, in “Abstract”, the expression “Destructive material testing…” should be modified to “Undestructive material testing…”; the statement “Digital Image Correlation (DIC)” should be revised to “digital image correlation (DIC)”. The authors are advised to consult with expert editors to remove language editing errors and minor technical typos within the paper’s manuscript. Consulting with the esteemed company (www.EditSprings.com ) is highly recommended for improving the English of the paper.

7.       The captions of the provided Tables should be placed at the top of the given Tables. Only in figures the captions are allowed to be located under the presented figure. Please revise them carefully.

Author Response

No.

Comments and Response

1

According to Table II, it seems that the numerically evaluated Elastic modulus values based on the natural frequency (i.e.,dynamic analysis) have higher relative errors with respect to those evaluated based on the static-based analysis.

Why? Please clarify within the paper manuscript.

The authors would like to clarify that the percentage errors shown on columns 4 and 7 of Table II pertain to the static deflection values and the natural frequencies of the beam respectively.  The values presented in these two columns are not the errors in the Young’s modulus E.

What the results show is that, for any given Young’s modulus E, the errors in the static deflections computed using theory and FEM are very small, between 0.01% and 0.015%.  For any given Young’s modulus E, the errors in the natural frequencies computed using theory and FEM are also very small, between 0.09% to 0.1%. 

The reason why the errors in the computed natural frequencies are higher than the computed static deflections is likely related to the different numerical techniques used to calculate these two quantities.  The static deflections are calculated using conventional stress/deformation-based techniques, while the natural frequencies are calculated using modal techniques involve eigenvalue analysis.  In this study, the eigenvalue analyses result in higher percentage errors compared to the stress/deformation analyses.  However, in both cases, the percentage errors are overall very small, and hence the FEM values can be considered to show good agreement with the theoretical values.   

2

Under what circumferences, we can use Eq. (1) for static analysis of beam-like structures? Please clarify carefully within the manuscript of the paper.

Eq. (1) is the governing equation for the flexural deflection of a slender beam.  It does not account for the occurrence of shear deformation during the flexural response, which may be present in the case of thick beams whose length-to-thickness ratios are smaller than 10 (as a rough rule of thumb).

The beam tested in this study has a high length-to-thickness ratio of 182, which is much higher than the value of 24 recommended in the ASTM standards [25] for such experiments.  Such a highly slender beam exhibits very negligible shear deformation during the flexural response, and hence the use of the Euler-Bernoulli beam theory (Eq.(1)) for calculating the transverse deflections and the natural frequencies in this study is valid.

3

For beams with length to height ratio lower than 5, the role of the shear deformation in the maximum deflection and natural frequencies of cantilevered beams become important.

Is the proposed approach by the authors would be also applicable to these special but crucial cases as well? How? Please explain with some detail.

The DIC measurements capture the real response of the physical structure, whose slenderness ratio may range from low values (< 5) to high values (> 20).  The effects of shear deformation, if any, will be captured in the DIC images.

For the numerical counterpart, the FEM model should be set up to replicate the physical structure as closely as possible.  This would require that a 3D model be set up and discretized into a fine mesh of solid elements that reflect the actual geometry and dimensions of the structure.  The finite element analysis of such a 3D model will capture the shear deformation effects, if any, that may be present in the response.  This is the approach that has been adopted in our study, in which the slender beam is simulated using 3D solid elements.

In the proposed approach, the use of 1-D line elements to model the beam in the finite element analysis is not wrong, but it is not encouraged.  Nevertheless, should such line elements be used to simulate a beam, then the formulation based on Timoshenko beam theory (which can account for shear deformation effects) should be used. 

4

In view of comment#3, the literature review on the application of shear deformable beam theories to the statics and dynamics of beam-like structures is somehow NOT complete! The authors are invited to discuss on the following works that employ Timoshenko and higher-order beam models for structural analysis of the beam-like structures:

The authors thank the reviewer for his recommendation of the publications related to Timoshenko and higher-order beam models.

As discussed in our response to Point 3, the proposed approach in our revised manuscript does not require the use of Timoshenko or higher-order beam models in the numerical analysis. It is recommended that the beam or any structure of interest be discretized using 3D solid elements, instead of line elements.  The use of a fine mesh of 3D solid elements can ensure that any shear deformation effects are properly and naturally captured in the model response.

5

Are there other effective non-destructive tests for measuring the elastic modulus of beams? Please clarify.

Yes, other non-destructive tests include impact excitation, flexural resonance, and several ultrasonic, resonance, and acoustic wave propagation and electronics techniques. Generally, these techniques require contact sensing. There are also new optical

non-contact technique using laser for Young’s modulus measurement. Nonetheless this method requires high cost equipment and complex setup.

6

There are some typos and badly constructed sentences that should be carefully corrected. For instance, in “Abstract”, the expression “Destructive material testing…” should be modified to “Undestructive material testing…”; the statement “Digital Image Correlation (DIC)” should be revised to “digital image correlation (DIC)”. The authors are advised to consult with expert editors toremove language editing errors and minor technical typos withinthe paper’s manuscript. Consulting with the esteemed company (www.EditSprings.com ) is highly recommended for improving the English of the paper.

Please accept our apologies for the earlier version of the manuscript which you reviewed.  We had mistakenly submitted an unedited version of the manuscript for review, which we later realize is full of language and grammatical errors.  We are sincerely sorry for the frustration you had experienced in trying to review and understand the contents of the unedited manuscript.  Your patience and understanding in reviewing the paper, despite the major language issues, is greatly appreciated.

The revised manuscript has been corrected and carefully checked to minimize such language issues.

7

The captions of the provided Tables should be placed at the top of the given Tables. Only in figures the captions are allowed to be located under the presented figure. Please revise them carefully

The authors thank the reviewers for pointing out this formatting rule for tables.  The table captions/titles have been moved so that they are located above the tables in the revised manuscript.

Round 2

Reviewer 2 Report

The made modifications are satisfactory.